## RESEARCH ARTICLE

# Alpha-synuclein overexpression does not cause vocalization deficits in a mouse model of parkinsonism

Brooke Rodgers and Allison Schaser*

## ABSTRACT

Voice deficits are common in Parkinson's disease (PD) and significantly impact quality of life by increasing stress, social isolation, and caregiver burden. However, despite this impact, there are currently no treatments that target the underlying pathophysiology of PD in the vocalization system. The goal of this study was to examine the effect of one possible underlying mechanism responsible for the complex voice deficits that exist in PD; overexpression of the protein alpha-synuclein. Results show that overexpression of alpha-synuclein, prior to the development of alpha-synuclein aggregate pathology, does not result in significant vocalization deficits. A small but statistically significant increase in the total number of complex vocalizations was found in mice overexpressing alpha-synuclein compared to wild-type mice, but there were no differences in complexity ratio or any of the other specific vocalization parameters tested. Results provide a critical foundational understanding of the impact of overexpression versus aggregation of alpha-synuclein on voice deficits in PD. Future work will focus on manipulation of alpha-synuclein aggregate pathology, and not overexpression alone, to reduce or eliminate the burden of PD specific voice disorders.

KEY WORDS: Voice, Ultrasonic vocalizations, Parkinsonism, Mouse model, Alpha-synuclein

## INTRODUCTION

Voice deficits commonly occur in Parkinson's disease (PD) and lead to significant impairments in quality of life (Ho et al., 1999; Miller, 2017). The neuropathological hallmarks of PD include development of alpha-synuclein aggregates in the central and peripheral nervous systems and progressive loss of dopaminergic neurons in the substantia nigra pars compacta (Seidel et al., 2015). However, it is currently unknown how alpha-synuclein pathology across the nervous system and central dopamine loss independently contribute to voice deficits in PD. The lack of mechanistic knowledge and the complex, variable nature of PD specific voice deficits make them inherently difficult to treat, which is highlighted by the controversy surrounding the inconsistent response of voice deficits to dopamine replacement therapy (Ho et al., 2008; Pinto et al., 2004). The inability of dopamine replacement therapy to effectively treat vocalization deficits leads to the hypothesis that some aspects of the vocal deficits in PD are not a result of the hallmark pathology in the striatal dopaminergic system but instead result from alpha-synuclein pathology in specific extra-striatal vocal communication areas. Consistent with this hypothesis, extra-striatal alpha-synuclein pathology has been shown in cranial nerves and brainstem nuclei that control voice in both human patients (Mu et al., 2012) and in mouse models of alpha-synuclein overexpression (Grant et al., 2014). If some aspects of PD specific voice deficits are caused by alpha-synuclein pathology in regions that influence voice and communication, including extra-striatal areas in the brainstem and cortico-bulbar regions, it would explain why voice deficits do not reliably respond to current gold standard therapies that focus on dopamine replacement strategies (Ma et al., 2020; Postuma and Berg, 2016; Seidel et al., 2015). To develop more robust and effective treatments for voice and communication deficits in PD, it is imperative that we determine the specific role of all possible mechanisms contributing to vocal deficits in PD.

Many transgenic mouse models have been developed to study alpha-synuclein pathology, but few studies have characterized vocalization behavior in these models. Alpha-synuclein overexpression has been modeled using wild-type (WT) and mutated (e.g. A53T, A30P) forms of the human *SNCA* gene with expression controlled by different gene promoters (e.g. mThy1, mPrion, PDGFb) (Chesselet and Richter, 2011; Winner et al., 2011). Gene multiplications of alpha-synuclein resulting in alpha-synuclein overexpression cause rare familial forms of PD and result in alpha-synuclein aggregation and pathology (Moore and Dawson, 2008). Similarly, alpha-synuclein overexpression has been shown to be useful to induce and study progressive alpha-synuclein pathology in animal models of PD (Chesselet and Richter, 2011). Most models of alpha-synuclein overexpression induce spontaneous alpha-synuclein aggregation and result in motor deficits similar to those observed in humans with PD (Chesselet and Richter, 2011; Chesselet et al., 2012; Grant et al., 2014; Winner et al., 2011). However, the effects of overexpression alone are rarely evaluated by researchers using these models, especially in studies of vocalization behavior. Grant and colleagues characterized progressive vocalization deficits in the Thy1-aSyn mouse model, which overexpresses the human WT form of alpha-synuclein via the Thy1 promoter. Compared to WT mice, the Thy1-aSyn mice were shown to have altered vocalization patterns as early as 2-3 months of age (Grant et al., 2014). The authors concluded that vocalization deficits in the Thy1-aSyn mice were attributed to alpha-synuclein pathology observed in the periaqueductal gray region, but the effect of overexpression alone was undetermined. Models of alpha-synuclein overexpression with spontaneous aggregation are important tools to study alpha-synuclein pathology; however, the spontaneous aggregation aspect of these models makes it difficult to study the effects of overexpression alone. In addition, the widespread pathology seen in these models makes it impossible to study the effects of alpha-synuclein pathology in discrete vocal communication areas. Given that alpha-synuclein overexpression is

Department of Speech, Language, and Hearing Sciences, Purdue University, West Lafayette, IN 47907, USA.

*Author for correspondence (aschaser@purdue.edu)

(iD) A.S., 0000-0002-4999-0565

present in rare cases of familial-PD and given that overexpression is often used to induce alpha-synuclein pathology in animal models of PD, it is necessary that we determine the effect of overexpression alone on vocalization. This information will allow us to best utilize the available animal models to develop targeted mechanistic treatments for PD specific vocalization deficits.

The purpose of this study was to directly examine the independent role of alpha-synuclein overexpression on mouse ultrasonic vocalizations (USVs). To do this we characterized USVs in the previously established A53T SynGFP mouse model, which was developed to study the inter- and intracellular spread of alpha-synuclein pathology in both the peripheral and central nervous system in combination with an alpha-synuclein fibril seeding approach (Schaser et al., 2020). The heterozygous A53T SynGFP mice overexpress a form of alpha-synuclein with the A53T mutation via the mouse prion protein promoter. The A53T mutation is a single amino-acid substitution in the *SCNA* gene at Alanine 53, which increases the probability of protein misfolding and is associated with familial early-onset PD (Giasson et al., 2002; Unger et al., 2006). Additionally, the strain was engineered to include a green fluorescent protein (GFP) tag on the mutant alpha-synuclein protein to aid genotyping and imaging. Unlike other alpha-synuclein overexpression mouse models (Giasson et al., 2002; Grant et al., 2014), the A53T SynGFP model does not develop spontaneous alpha-synuclein pathology over time and does not develop PD specific motor deficits with advanced age (Schaser et al., 2020). In Schaser et al. (2020), A53T SynGFP transgenic mice were injected with either pre-formed fibrils (PFFs) of aggregated alpha-synuclein or monomeric alpha-synuclein. Only mice treated with PFFs developed alpha-synuclein aggregate pathology; mice that received monomeric alpha-synuclein did not show any alpha-synuclein pathology 4- and 8-months post-injection (Schaser et al., 2020). In this study, we used the A53T SynGFP mouse model to determine if alpha-synuclein overexpression, prior to the onset of aggregate pathology, results in vocalization deficits.

## RESULTS

USVs were elicited from a total of 29 A53T SynGFP and WT mice of both sexes using an adapted mating paradigm. For our purposes, only USVs in the 50 kHz range, longer than 10 ms were included for analysis. USVs with only one acoustic element (flat, up/down, and U-shaped USVs) were considered simple vocalizations, and USVs with multiple acoustic elements (trailing, step up/down, and complex USVs) were considered complex vocalizations for analysis. Fig. 1 includes example spectrograms of simple and complex USVs produced by a male A53T SynGFP mouse. The USV characteristics of interest were the total USV count, complexity ratio (calculated as the number of simple USVs divided by the number of complex USVs), duration (ms), frequency range (kHz) (calculated as Fmax – Fmin), and the maximum intensity (dB).

Over 19,000 vocalizations were identified by the USV identification software and included for statistical analysis. Of the vocalizations identified, 14,879 USVs were classified as simple and 4245 USVs were classified as complex. Two-way analysis of variance (ANOVA) models were used to test the effects of sex and genotype on each USV characteristic. Due to the Poisson distribution, total USV counts were analyzed using two-way analysis of deviance models. The mean values for each genotype and sex group are shown in Table 1, and results of the statistical tests are shown in Table 2. Plots showing the genotype group averages are shown in Fig. 2. All animals produced more simple than complex USVs and had complexity ratios greater than 1. Across both groups, complex vocalizations had greater durations, frequency ranges, and maximum intensities than simple vocalizations, on average. There were no statistically significant effects due to sex for any of the acoustic variables for either USV complexity type: total USV count [simple: $Pr(>\chi^2)=0.179$; complex: $Pr(>\chi^2)=0.252$], complexity ratio ($P=0.702$), duration (simple: $P=0.314$; complex: $P=0.943$), frequency range (simple: $P=0.214$; complex: $P=0.935$), and maximum intensity (simple: $P=0.406$; complex: $P=0.0524$). There were no statistically significant effects

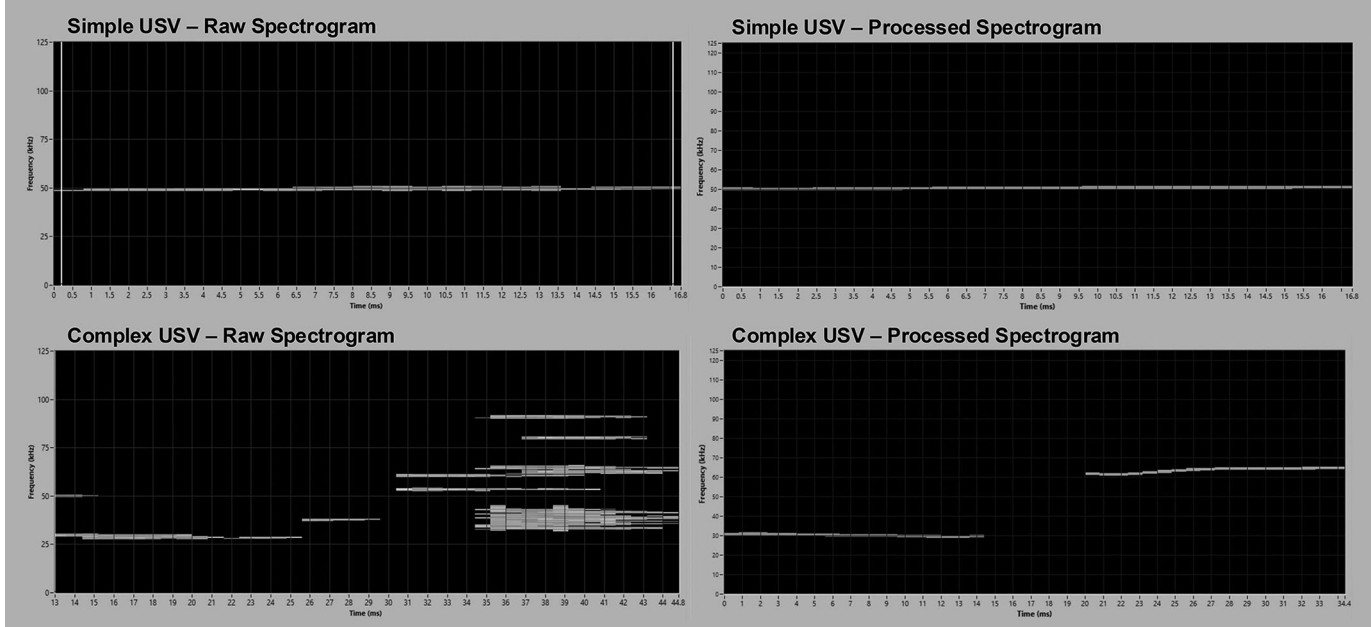

Fig. 1. Example spectrograms of simple and complex USVs. Representative images showing example spectrograms of a simple (top row) and complex (bottom row) USV produced by a male A53T SynGFP mouse. Both raw spectrograms (left column) and spectrograms after processing (right column) with the Sonotrack software are shown. The USV duration is shown on the x-axis, the USV frequency is shown on the y-axis, and the shade of gray in the raw spectrograms indicates the intensity of the USV.

**Table 1. Sex and genotype group averages (mean±s.e.m.) for each USV characteristic calculated for the two genotype groups (A53T SynGFP+ and WT) and averaged across sex**

| | Simple USVs | | | | Complex USVs | | | |
|---|---|---|---|---|---|---|---|---|
| | A53T SynGFP+ | | WT | | A53T SynGFP+ | | WT | |
| Acoustic characteristic | Male | Female | Male | Female | Male | Female | Male | Female |
| Average total count | 511±8.54 | 508±7.52 | 503±8.48 | 534±9.43 | 148±4.60 | 153±4.12 | 139±4.46 | 143±4.89 |
| Average complexity ratio[1] | 3.72±0.366 | 3.83±0.322 | 3.97±0.366 | 4.21±0.395 | | | | |
| Average duration (ms) | 13.8±0.186 | 13.7±0.164 | 13.6±0.186 | 13.2±0.201 | 41.4±1.85 | 41.1±1.63 | 40.6±1.85 | 40.5±2.00 |
| Average frequency range (kHz) | 3.21±0.221 | 2.85±0.195 | 2.84±0.221 | 2.60±0.238 | 23.1±0.976 | 22.9±0.861 | 22.6±0.976 | 22.9±1.054 |
| Average maximum intensity (dB) | −33.9±1.62 | −35.6±1.43 | −35.1±1.62 | −36.3±1.75 | −13.5±1.19 | −16.3±1.05 | −12.6±1.19 | −14.1±1.28 |

[1]Complexity ratio is the quotient of the number of simple and complex USVs produced (Ratio=# simple USVs/# complex USVs).

due to genotype for total simple USV count [$Pr(>\chi^2)$=0.279], complexity ratio ($P$=0.392), duration (simple: $P$=0.107; complex: $P$=0.712), frequency range (simple: $P$=0.167; complex: $P$=0.816), and maximum intensity (simple: $P$=0.57; complex: $P$=0.194). There was a statistically significant effect due to genotype for the total number of complex vocalizations produced [$Pr(>\chi^2)$=0.0429]. Over the five recording days, animals overexpressing alpha-synuclein (A53T SynGFP+) produced nine more complex USVs (or <2 more complex USVs per day) than WT animals on average.

## DISCUSSION

Results of this study show that overexpression of alpha-synuclein alone does not result in significant vocalization deficits in the specific transgenic model used in this study. Using the A53T SynGFP mouse model, which overexpresses alpha-synuclein without developing spontaneous alpha-synuclein aggregate pathology, we were able to directly test the influence of alpha-synuclein overexpression alone on multiple USV parameters. We found that there was a small but statistically significant increase in

**Table 2. Results of two-way ANOVA statistical tests**

| | Simple USVs | Complex USVs |
|---|---|---|
| **Acoustic variable** | | |
| • Analysis of deviance factor | Dev (df, df); $Pr(>\chi^2)$ | Dev (df, df); $Pr(>\chi^2)$ |
| Total USV count | | |
| • Sex | 323 (1,27); 0.179 | 205 (1,27); 0.252 |
| • Genotype | 321 (1,26); 0.279 | 201 (1,26); **0.0429***|
| • Sex:genotype | 318 (1,25); 0.0536 | 201 (1,25); 0.984 |
| | Simple USVs | Complex USVs |
| **Acoustic variable** | | |
| • ANOVA factor | $F_{(1,25)}$; $P$-value | $F_{(1,25)}$; $P$-value |
| Complexity ratio[1] | | |
| • Sex | 0.149; 0.702 | |
| • Genotype | 0.760; 0.392 | |
| • Sex:genotype | 0.0261; 0.873 | |
| Duration | | |
| • Sex | 1.06; 0.314 | 0.00530; 0.943 |
| • Genotype | 2.80; 0.107 | 0.140; 0.712 |
| • Sex:genotype | 0.892; 0.354 | 0.00630; 0.937 |
| Frequency range | | |
| • Sex | 1.63; 0.214 | 0.00670; 0.935 |
| • Genotype | 2.02; 0.167 | 0.0552; 0.816 |
| • Sex:genotype | 0.0668; 0.798 | 0.0575; 0.813 |
| Maximum intensity | | |
| • Sex | 0.715; 0.406 | 4.15; 0.0524 |
| • Genotype | 0.331; 0.570 | 1.78; 0.194 |
| • Sex:genotype | 0.0166; 0.899 | 0.335; 0.568 |

[1]Complexity ratio is the quotient of the number of simple and complex USVs produced (Ratio=# simple USVs/# complex USVs).
*Denotes statistically significant effect $Pr(>\chi^2)$<0.05.

the total number of complex vocalizations produced by the A53T SynGFP+ mice compared to the WT mice in this study. However, the overall increase of nine total complex USVs across 5 days does not appear to represent a clinically meaningful change in vocalization and does not represent a vocalization deficit typically seen in clinical PD (Darley et al., 1968; Ho et al., 1999; Plowman-Prine et al., 2009). People with PD have reported initial changes in voice quality followed by reductions in volume and prosody as the disease progresses (Ho et al., 1999). Vocalization deficits observed in the Thy1-aSyn mouse model included reduced USV rate, duration, and intensity (Grant et al., 2014). We expect mice with vocalization deficits caused by alpha-synuclein pathology to produce fewer USVs, a higher proportion of simple USVs, and reduced intensity. The A53T SynGFP+ mice, however, were shown to produce slightly more complex USVs than WT littermates.

All other USV parameters tested in the study were not different in A53T SynGFP+ mice compared to WT mice, indicating that overexpression of alpha-synuclein alone does not result in significant vocal deficits in this specific mouse model. This result is different from a previous study that examined vocalization in mice overexpressing alpha-synuclein (Grant et al., 2014). The Thy1-aSyn mice used in the previous study demonstrated early, progressive vocalization deficits compared to WT controls. However, even at the early time points tested in the Grant et al. (2014) study, vocalization deficits were linked to alpha-synuclein aggregation in the vocalization system, specifically in the periaqueductal gray (Grant et al., 2014). The need for alpha-synuclein aggregation to be present in specific areas to induce vocalization deficits is supported by previous work in a rat PFF injection model which has shown that vocalization deficits do occur after PFF injection into the striatum, but these deficits only occurred once widespread pathology had developed over a 6-month period and did not appear to affect vocalization quality (bandwidth and intensity) (Paumier et al., 2015). This finding suggests a role for extra-striatal alpha-synuclein pathology in the overall scope of PD specific vocalization deficits. The results of Grant et al. (2014), Paumier et al. (2015), and the results reported here indicate that overexpression of alpha-synuclein alone is not sufficient to cause vocalization deficits, and suggests the need for underlying alpha-synuclein aggregate pathology to cause vocalization deficits.

Future studies will benefit from the use of the A53T SynGFP transgenic mouse model tested here because it eliminates the concern that overexpression of alpha-synuclein alone influences USVs (Schaser et al., 2020). Using this mouse model, we can control the location and timing of the development of alpha-synuclein aggregate pathology by taking advantage of additional methods to induce alpha-synuclein aggregate pathology. Alpha-synuclein pathology propagation can be induced by the exogenous application of small, *in vitro*-generated alpha-synuclein PFFs. Previous research has

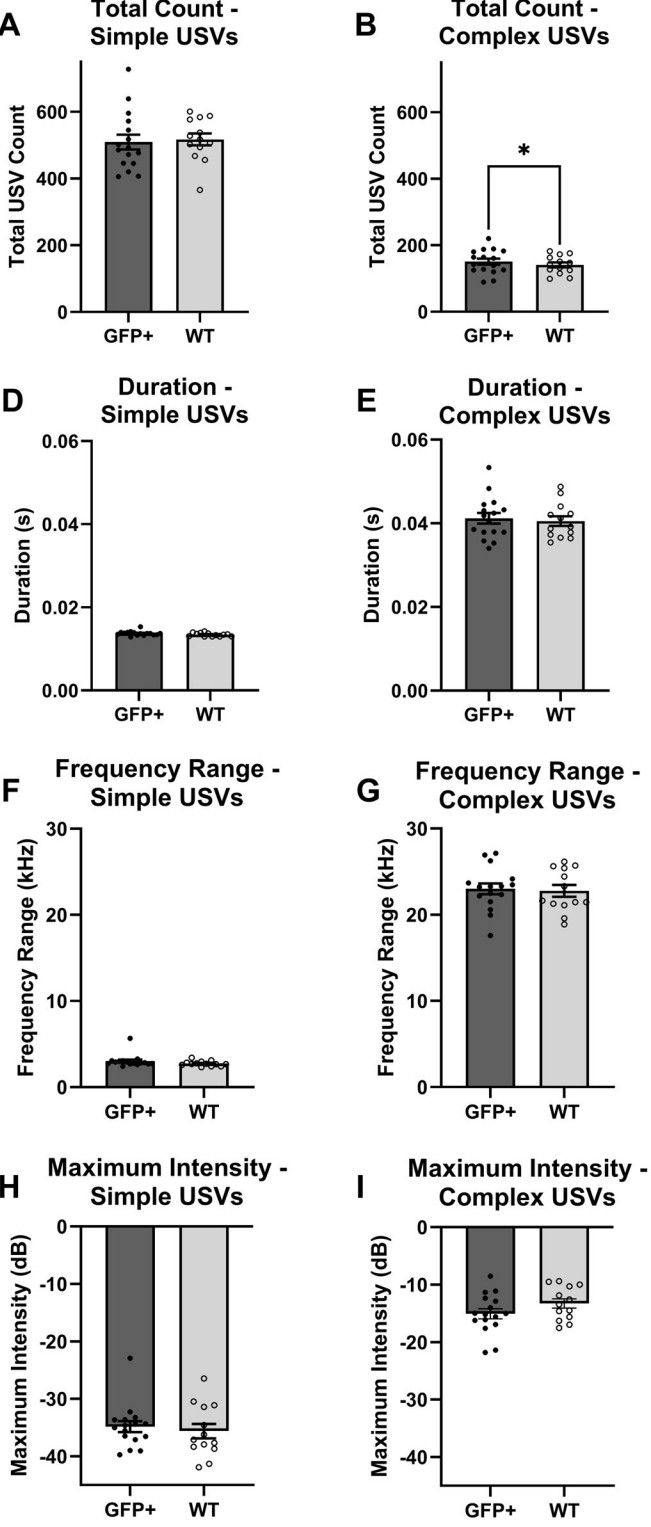

**Fig. 2. USV results averaged over each genotype group (GFP+, A53T SynGFP+; WT, wild type), pooled across sex.** Results for simple (left column) and complex (right column) USVs were analyzed and plotted separately as total USV count (A,B), duration (D,E), frequency range (F,G), and maximum intensity (H,I). Group averages for USV complexity ratio are shown in C. Error bars represent the standard error of the mean value (s.e.m.) and points depict values for individual animals. The asterisk above the bars in plot B indicate a statistically significant difference between the group means observed in the analysis of deviance model [$Pr(>\chi^2)<0.05$].

shown that the application of PFFs causes aggregation of endogenous alpha-synuclein and resultant pathology first at the site of injection and then in connected brain regions in a time dependent manner (Luk et al., 2009; Osterberg et al., 2015; Sacino et al., 2014; Schaser et al., 2020). Future studies should employ the current A53T SynGFP mouse model and PFF method to determine the effects of alpha-synuclein pathology in discrete areas of the vocal communication system.

The specificity and inducibility of the A53T SynGFP mouse model paired with the PFF method make it an ideal system to examine alpha-synuclein aggregate pathology in striatal versus extra-striatal components of the vocal communication system. Specifically, it will be important to induce alpha-synuclein pathology in areas along the vocal communication neuro-axis – including the vocal part of the periaqueductal gray and the layer V pyramidal neurons in the singing activated region of M1 (Arriaga et al., 2012). The A53T SynGFP

mouse model is ideal because it allows for rapid induction of pathology exclusively in one area while preserving the health of other areas (Schaser et al., 2020). This also allows for direct examination of the effect of alpha-synuclein pathology in discrete areas on vocal communication behavior, without the need to control for the additional effects of alpha-synuclein overexpression. Additionally, the GFP tag aids in genotyping mice and imaging alpha-synuclein pathology. Future studies will use the A53T SynGFP mouse model in combination with the PFF method to elucidate the effect of alpha-synuclein pathology in discrete vocal communication areas on specific acoustic parameters. The overall goal of future work using this mouse model and the PFF method will be to determine the underlying cause of vocalization deficits in PD and to develop targeted treatments that address the variable and complex vocalization deficits that occur in PD.

The results of this study lay the groundwork for the development of a mouse model that allows for manipulation of alpha-synuclein within discrete areas of the vocal communication system and enables the analysis of vocalizations in a relevant behavioral manner, without the need to control for the baseline level of alpha-synuclein overexpression. Our results provide a critical and necessary foundational understanding of the role of alpha-synuclein overexpression in animal models of PD specific voice deficits. The results of this study are critical to our future work using the A53T SynGFP overexpression model in combination with the PFF method to study aggregation and spread of alpha-synuclein pathology as one of the underlying mechanisms that lead to vocalization deficits in PD. Our data strongly suggest that this mouse model represents an ideal baseline system that can be paired with the PFF method to test important fundamental questions related to how alpha-synuclein pathology propagation occurs within the vocal communication system and results in PD specific voice and communication disorders.

## MATERIALS AND METHODS
### Animals and ethics statement
The procedures included in this study were performed in accordance with a protocol (IP00001084) approved by the Oregon Health & Science University (OHSU) Institutional Animal Care and Use Committee. Animals were housed in a 12-h light–dark cycle, temperature- and humidity-controlled vivarium at OHSU, and maintained under *ad libitum* food and water diet. All experiments were performed in accordance with the relevant guidelines and regulations, and every effort was made to minimize suffering and the number of animals used.

All experiments in this study were carried out using a previously characterized alpha-synuclein transgenic mouse model, known as the A53T SynGFP mouse model (Schaser et al., 2020). Animals used in this study were bred from the same colony used for the experiments reported in Schaser et al. (2020). Animals were bred at OHSU by mating male heterozygous A53T SynGFP+ mice with WT females. Breeder mice were housed in pairs in conventional solid-bottom cages and litters were weaned at 21 days of age. At weaning, mouse pups were genotyped using an ultraviolet flashlight (PN 550-1009; Light & Motion, currently unavailable) and separated by sex. Mice were housed in single-sex groups of three to five mice in conventional solid-bottom cages until the time of recording. A total of 29 male and female mice, aged 3-6 months, were used in this study. Sixteen transgenic (A53T SynGFP+) (seven male and nine female) mice and 13 WT (seven male and six female) littermate control mice were used to determine if alpha-synuclein overexpression alone can cause vocalization deficits.

### USV elicitation procedure
USVs were elicited using an adapted existing mating paradigm (Chabout et al., 2017). Mice underwent five consecutive days of habituation followed by five consecutive days of recording. Mice underwent the same procedures during habituation and recording, but mice were not exposed to the stimulus or recorded on habituation days. USVs were recorded from individual mice placed into a custom sound-attenuated chamber fit with an ultrasonic recording system (Avisoft, Glienicke, Germany). Vocalizations were induced by stimulating each mouse with urine from an unfamiliar C57BL/6j mouse. Mouse urine was purchased (PN MSE01URINEPNN; BioIVT, Woodbury, NY, USA) and introduced into the testing cage via a cotton swab. Following exposure to the urine, the scent elicited USVs from the mice. 5 min of USVs were recorded from each animal each day.

### Recording and USV identification
USVs were recorded using an ultrasonic recording system (Avisoft, Glienicke, Germany) with appropriate wide frequency response range (10-180 kHz) and the capability of producing spectrograms in real time. USVs were identified using the Sonotrack software package (Metris, Hoofddorp, The Netherlands). After the final recording session, all recording files were processed automatically in Sonotrack using a custom set of parameters that identified USVs within a specified frequency range (25-100 kHz) and above a threshold duration (10 ms). The identified USVs were further classified by complexity based on acoustic parameters informed by previously published studies and the Metris categorization guidelines (Metris, 2017). USVs with only one acoustic component (flat, up/down, and chevron type vocalizations) were considered 'simple' USVs, and USVs with multiple acoustic components (trailing, step up/down, split up/down, and complex type vocalizations) were considered 'complex' USVs for our statistical analyses. Fig. 1 includes example spectrograms of simple and complex USVs produced by a male A53T SynGFP mouse.

### Statistical analysis
The USV statistical analyses were performed using R Statistical Software (v4.3.0; R Core Team, Vienna, Austria). After USV identification using Sonotrack, the output spreadsheets were concatenated to create a complete dataset including all USVs produced during each recording session. The variables of interest for statistical analysis included: 1) total USV counts, 2) the ratio of the number 'simple' to 'complex' USVs produced, 3) the duration of USVs in seconds (s), 4) the frequency range of USVs in kilohertz (kHz), and 5) the maximum intensity of USVs in decibels (dB). The total USV count was calculated by summing the number of USVs produced by each animal over the five recording days. The complexity ratio was calculated by dividing the number of simple USVs produced by the number of complex USVs produced by each animal each day (Ratio=# simple USVs/# complex USVs). The average complexity ratio was then calculated over the five recording days for each animal to use for statistical analysis. For the USV duration, frequency range, and maximum intensity, all the USVs produced were pooled over the five recording days to determine the average value for each animal. The average USV duration, frequency range, and maximum intensity were used for statistical analysis. Each variable was analyzed using a two-way ANOVA model with the factors of sex (male or female) and genotype (WT or A53T SynGFP+), except for total USV counts. Due to the Poisson distribution, total USV counts were analyzed using two-way analysis of deviance models applied to generalized linear models. Two-way interaction effects were included in the models. These models were used to evaluate any differences in the USVs produced by the mice due to genotype and/or sex at baseline. Group mean differences were considered statistically significant when $P<0.05$. For total USV counts, group mean differences were considered statistically significant when $Pr(>\chi^2)<0.05$.

Analyses were stratified between simple and complex USVs. Analyses were separated because of the acoustic parameters used to classify the USVs as simple or complex. Separate datasets were created to only include USVs classified as simple or complex, and the same statistical tests were performed on both datasets.

Estrous stage was not measured in the female mice during the recording period, but each stage of the 4-5-day cycles in the mice was captured over the five recording days (Zeng et al., 2023). We did not control for estrous stage in our statistical models but instead averaged the acoustic variables over the 5 days to remove effects due to estrous stage.

## Acknowledgements

The authors would like to acknowledge V. Unni, the OHSU animal care staff, the OHSU Transgenic Mouse Model Core, The Purdue Institute for Integrative Neuroscience, and the Purdue Animal Behavior Core for their assistance and support of the study. They would also like to acknowledge M. Norton and B. Hernandez for their assistance with data collection and analysis for this study.

## Competing interests

The authors declare no competing or financial interests.

## Author contributions

Conceptualization: A.S.; Data curation: A.S., B.R.; Formal analysis: B.R.; Funding acquisition: A.S.; Methodology: A.S., B.R.; Project administration: A.S.; Visualization: B.R.; Writing – original draft: A.S., B.R.; Writing – review & editing: A.S., B.R.

## Funding

This work was supported in part by the Collins Medical Trust Grants (A.S.) and an American Speech-Language Hearing Foundation New Investigators Research Grant (A.S.). Open Access funding provided by Purdue University. Deposited in PMC for immediate release.

## Data and resource availability

All relevant USV data files, analysis script, and supporting images are publicly available through the Purdue University Research Repository (DOI: 10.4231/7V1E-7163; https://purr.purdue.edu/publications/4583/1).

## Peer review history

The peer review history is available online at https://journals.biologists.com/bio/lookup/doi/10.1242/bio.062120.reviewer-comments.pdf

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
