## [Peer Review File · Biology Open]

Alpha-synuclein overexpression without vocalization deficits in a mouse model of parkinsonism

Brooke Rodgers and Allison J. Schaser

DOI: 10.1242/bio.062120

Editor: Sandhya Koushika

Review timeline

Original submission:	18 June 2025
Editorial decision:	26 June 2025
First revision received:	28 July 2025
Accepted:	4 August 2025

Original submission

First decision letter

MS Title: Alpha-synuclein overexpression without vocalization deficits in a mouse model of parkinsonism.

Authors: Allison J Schaser; Brooke Rodgers

Article Type: Research Article

I have now reached a decision on the above manuscript.

The reviewer reports are shown at the bottom of this email or can be accessed, together with a copy of this decision letter, by going to:

As you will see, the reviewers raised a number of substantial criticisms that prevent me from accepting the paper at this stage.

They suggest, however, that a revised version might prove acceptable, if you can address their concerns. If you think that you can deal satisfactorily with the criticisms on revision, I would be pleased to see a revised manuscript. We would then return it to the reviewers.

At this stage, we also ask you to ensure your manuscript complies with our formatting guidelines. Provided you are able to fully address the referees' comments, we are positive about publication of your paper (we accept over 95% of revision submissions) and therefore hope you won't mind any extra work involved in reformatting your manuscript at this point.

Please ensure that you clearly highlight all changes made in the revised manuscript. Please avoid using 'Tracked changes' in Word files as these are lost in PDF conversion.

I should be grateful if you would also provide a point-by-point response detailing how you have dealt with the points raised by the reviewers in the 'Response to Reviewers' box. Please attend to all of the reviewers' comments. If you do not agree with any of their criticisms or suggestions please explain clearly why this is so.

Reviewer 1

Overall comments:

In their study, Rodgers and Schaser test the hypothesis that overexpression of alpha-synuclein contributes to voice deficits in a mouse model. This question is important in its field because voice deficits are a significant area of loss of life quality in people with Parkinson's disease (PD), and because alpha-synuclein aggregation is a symptom of PD that may be linked to vocal deficits. Previous work to understand the mechanistic cause of voice deficits in PD has focused on dopaminergic pathways but with no success. Thus, new mechanisms should be explored and a potential promising mechanism for alpha-synuclein has not received research attention regarding voice deficits to date. The authors found that mice overexpressing alpha-synuclein had a small (likely biologically insignificant) increase in complex vocalizations, but no other changes were observed between mutant and wild-type mice. The authors suggest that this work will provide a floor for future work exploring the impact of alpha-synuclein on PD symptoms in animal models. The paper is clearly written, and an appropriate justification of the study is provided. Presenting the data in both table and graph format may be redundant but overall, the data presentation and discussion is also clear and robust. I offer a few comments for the authors to consider:

Major Comments:

1. It is not clear to me why this study is an advance beyond the work of the Grant et al 2014 paper cited. The two studies used different transgenic mice, but it seems that both mutants were designed to have overexpression of alpha-synuclein. The previous study was more robust from an experimental perspective, and included analysis at more ages, etc. The previous study noted vocal deficits in several measures and at different ages and an impact of the mutation, but the current study does not replicate these findings. The authors need to clearly explain why their study is not redundant to the previous study. Is this simply a test of a different mouse model of alpha synuclein overexpression? Is it important because this model does not lead to aggregate formation? If not, why is it an improvement on the previous mutant model? Why are the results different? This background is important but not clearly established. Please define the advance of the present paper relative to the previous paper and discuss as appropriate in the intro and discussion of your revised manuscript.
2. The authors state that their mouse model doesn't form alpha-synuclein aggregates. Please provide evidence of this or a citation to a study that demonstrates this in this mutant model.
3. The Grant et al study found deficits at different ages in knockout mice, but not all ages studied. The authors should consider the age chosen to examine vocal deficits in the present study and explain why this age was chosen. Previous studies found changes in 6-7 month old mice but not in 2-3 month old mice. The age of the mice should be considered in the statistical analysis to see if there is any impact that is masked by pooling the data across the given age range of 3-6 months in this study. Please discuss whether different results would be expected from older or younger knock-out mice.
4. The authors recorded an impressive number of vocalizations for their analysis and indicated that they obtained recordings from both sexes in control and mutant mice. However, the analysis of this data by sex is not presented in the paper. Please consider breaking down your results by sex.
5. Would it be possible to include examples of complex vs. simple sound waveforms as a figure in this paper? (Similar to Fig. 1 in Grant et al 2014).
6. Please provide stats tables with your full ANOVA results/

Minor Comments:

1. Please indicate your statistical test in the figure and table legends.
2. Please provide a statement regarding animal ethics, including your animal use protocol number and information about the relevant governing bodies.

3. Abbreviations in the methods are defined that had already been defined in the introduction. Please revise for consistency.

Matthew Pamenter

Reviewer 2

Voice deficits are commonly observed in patients with Parkinson's disease (PD). A significant amount of evidence suggests that the aggregation of alpha-synuclein participates in the pathophysiology of PD. The objective of this study by Rodgers and Schaser was to further explore the relationship between alpha-synuclein expression and vocalization defects. They specifically determined if the overexpression of alpha-synuclein (before aggregation) in a PD mouse model was sufficient to cause voice defects. Results show that alpha-synuclein overexpression alone is not sufficient to cause significant vocalization deficits. These data suggest that further work should focus on alpha synuclein aggregation to develop therapeutic approaches that target voice deficits in PD.

The short study concludes with a negative result, demonstrating that alpha-synuclein overexpression in the A53T SynGFP mouse model (which does not spontaneously form aggregates over time) does not cause significant vocalization defects compared to wild-type mice. This finding is supported by published data showing that alpha-synuclein aggregates are observed in extra-striatal voice communication areas, suggesting a connection with voice defects seen in PD. While the overexpression data is interesting, the overall study could be improved in several ways.

Major comments:

1) The most obvious missing piece of data is what should serve as the positive control. The findings would be significantly strengthened by including A53T SynGFP mice that have been injected with pre-formed fibrils to induce alpha-synuclein aggregation. The prediction would be that this would support the author's hypothesis that aggregation is required to cause vocalization defects. Moreover, it would significantly strengthen the conclusion that overexpression alone is insufficient.

2) Even though this has presumably been published elsewhere, overexpression in the appropriate brain regions should be confirmed by immunohistochemistry. If this is not possible, broad overexpression in the brain could be confirmed by qPCR or Western blotting.

3) The data showing the effects are not due to sex (lines 95-99) do not seem to be included in a table or figure.

4) How the Thy1-aSyn mouse impacts alpha-synuclein pathology (line 132) and how it differs from A53T SynGFP needs to be better explained. Moreover, how findings with this model suggest that overexpression alone is insufficient to cause vocalization defects needs to be clarified (as stated in lines 136-139).

5) The authors should be careful concluding that this current study supports the hypothesis that vocalization deficits require the presence of alpha-synuclein pathology. This can be found throughout the text but is highlighted in the last paragraph of the discussion (lines 179-189). The current study does not show that the A53T SynGFP mouse will be a useful model for studying the relationship between alpha-synuclein aggregation and PD-specific voice and communication disorders. In its current form, this study only suggests that overexpression alone is insufficient. In contrast, a positive result in the additional experiment proposed in major comment #1 above would support this conclusion.

Minor comments:

1) Given that the difference was statistically significant, the authors could expand on their discussion of complex vocalizations (lines 122-128). Even if it does not mimic vocalization changes in PD patients.

2) The introduction could include a short description of how alpha-synuclein dysregulation contributes to PD pathology. At the least, a statement regarding the relationship between alpha-synuclein overexpression and PD should be included.

3) Make sure the source of all critical purchased materials is included in the methods to facilitate the reproducibility of experiments.

Reviewer's Responses to Questions

Experimental quality

Does each figure have the proper controls?

If 'No', please indicate reasons in Comments for Author box below.

Reviewer #1:

- Yes

Reviewer #2:

- No

Were the data analyzed using appropriate statistical tests?

If 'No', please indicate reasons in Comments for Author box below.

Reviewer #1:

- Yes

Reviewer #2:

- Yes

Reproducibility

Were experiments performed using adequate number of biological replicates?

If 'No', please indicate reasons in Comments for Author box below.

Reviewer #1:

- Yes

Reviewer #2:

- Yes

Does the methods section provide sufficient detail to permit reproducibility?

If 'No', please indicate reasons in Comments for Author box below.

Reviewer #1:

- Yes

Reviewer #2:

- No

Completeness

Are the manuscript's conclusions supported by the data?

If 'No', please indicate reasons in Comments for Author box below.

Reviewer #1:

- Yes

Reviewer #2:

- Yes

Scholarship

Do the authors cite and discuss the merits of data that would argue for and against their conclusion?

If 'No', please indicate reasons in Comments for Author box below.

Reviewer #1:

- Yes

Reviewer #2:

- Yes

Does the manuscript title & abstract accurately reflect the contents of the manuscript, without hyperbole?

If 'No', please indicate reasons in Comments for Author box below.

Reviewer #1:

- Yes

Reviewer #2:

- Yes

First revision

Author response to reviewers' comments

Reviewer 1. It is not clear to me why this study is an advance beyond the work of the Grant et al 2014 paper cited. The two studies used different transgenic mice, but it seems that both mutants were designed to have overexpression of alpha-synuclein. The previous study was more robust from an experimental perspective, and included analysis at more ages, etc. The previous study noted vocal deficits in several measures and at different ages and an impact of the mutation, but the current study does not replicate these findings. The authors need to clearly explain why their study is not redundant to the previous study. Is this simply a test of a different mouse model of alpha synuclein overexpression? Is it important because this model does not lead to aggregate formation? If not, why is it an improvement on the previous mutant model? Why are the results different? This background is important but not clearly established. Please define the advance of the present paper relative to the previous paper and discuss as appropriate in the intro and discussion of your revised manuscript.

Response: We thank the reviewer for these questions and suggestions. We have added additional information to the Introduction from the alpha-synuclein overexpression literature to distinguish our study. The most important aspect of our study is the fact that this model does not lead to spontaneous aggregate formation. Previous studies using overexpression models with spontaneous protein aggregation are unable to determine the effect of overexpression alone. This study is the

only study, to our knowledge, that separates overexpression alone, from overexpression in the context of protein aggregation.

Reviewer 1. The authors state that their mouse model doesn't form alpha-synuclein aggregates. Please provide evidence of this or a citation to a study that demonstrates this in this mutant model.

Response: Thank you for this suggestion. Schaser *et al.* (2020) includes this information. We have added more information to the Introduction to highlight this study and the A53T SynGFP model to clarify this point.

Reviewer 1. The Grant et al study found deficits at different ages in knockout mice, but not all ages studied. The authors should consider the age chosen to examine vocal deficits in the present study and explain why this age was chosen. Previous studies found changes in 6-7 month old mice but not in 2-3 month old mice. The age of the mice should be considered in the statistical analysis to see if there is any impact that is masked by pooling the data across the given age range of 3-6 months in this study. Please discuss whether different results would be expected from older or younger knock-out mice.

Response: Thank you for this suggestion, we agree that age is an important factor in the Grant et al study because in their mouse model spontaneous aggregation occurs overtime in different locations throughout the neuroaxis. We did not include enough animals of different ages to analyze the statistical effect of age in our study. However, previous work (Schaser et al. (2020)) confirms the absence of pathology at the ages used in this study. We agree that age may be important to consider regardless of pathology, since there are known age-related vocalization changes, and will plan future studies to include age as a factor in our analysis.

Reviewer 1. Please provide stats tables with your full ANOVA results.

Response: Thank you for this suggestion. We have added an additional table showing the results of our ANOVA model (Table 2).

Reviewer 1. The authors recorded an impressive number of vocalizations for their analysis and indicated that they obtained recordings from both sexes in control and mutant mice. However, the analysis of this data by sex is not presented in the paper. Please consider breaking down your results by sex.

Response: Thank you for this suggestion. Sex was included as a factor in our statistical analysis, but no differences due to sex were observed for any of the acoustic variables tested. These results are included with the ANOVA results table added to the Results section. Because no differences due to sex were observed, we chose to only show genotype in the results Figure 2.

Reviewer 1. Would it be possible to include examples of complex vs. simple sound waveforms as a figure in this paper? (Similar to Fig. 1 in Grant et al 2014).

Response: Thank you for this suggestion. We have added spectrogram images showing examples of simple and complex USVs in Figure 1.

Reviewer 1. Please indicate your statistical test in the figure and table legends.

Response: Thank you for this suggestion. We have updated our figure and table legends accordingly.

Reviewer 1. Please provide a statement regarding animal ethics, including your animal use protocol number and information about the relevant governing bodies.

Response: Thank you for this comment. We have included additional information regarding animal ethics and use and have included the protocol number that was active at the time of the study.

Reviewer 1. Abbreviations in the methods are defined that had already been defined in the introduction. Please revise for consistency.

Response: Thank you for this comment. We have revised the manuscript to use all abbreviations consistently throughout.

Reviewer 2. The most obvious missing piece of data is what should serve as the positive control. The findings would be significantly strengthened by including A53T SynGFP mice that have been injected with pre-formed fibrils to induce alpha-synuclein aggregation. The prediction would be that this would support the author's hypothesis that aggregation is required to cause vocalization defects. Moreover, it would significantly strengthen the conclusion that overexpression alone is insufficient.

Response: Thank you for this suggestion. Though we did not include USV recordings from PFF-treated transgenic animals in this study, we are including WT and transgenic animals treated with PFFs in our ongoing experiments and will report the results of those experiments in future publications. We agree that this information is important to the argument that alpha-synuclein pathology is required to induce vocalization deficits and have tempered our conclusions with this in mind.

Reviewer 2. Even though this has presumably been published elsewhere, overexpression in the appropriate brain regions should be confirmed by immunohistochemistry. If this is not possible, broad overexpression in the brain could be confirmed by qPCR or Western blotting.

Response: Thank you for this suggestion. The animals used in this study were from the same colony of animals used in Schaser *et al.* (2020), which included confirmation of overexpression and absence of pathology by immunohistochemistry in untreated A53T SynGFP animals of the same age range. We have clarified this information in the methods section.

Reviewer 2. The data showing the effects are not due to sex (lines 95-99) do not seem to be included in a table or figure.

Response: Thank you for this comment. We originally did not include statistical results related to sex because no differences were observed. We have edited our results table and added an additional table to show the full results of our statistical analysis, including results related to sex. Because there were no effects due to sex, we chose to only show genotype group averages in Figure 2.

Reviewer 2. How the Thy1-aSyn mouse impacts alpha-synuclein pathology (line 132) and how it differs from A53T SynGFP needs to be better explained. Moreover, how findings with this model suggest that overexpression alone is insufficient to cause vocalization defects needs to be clarified (as stated in lines 136-139).

Response: Thank you for this suggestion. We have included more information about other transgenic alpha-synuclein models and how the A53T SynGFP model differs from previous models in the introduction.

Reviewer 2. The authors should be careful concluding that this current study supports the hypothesis that vocalization deficits require the presence of alpha-synuclein pathology. This can be found throughout the text but is highlighted in the last paragraph of the discussion (lines 179-189). The current study does not show that the A53T SynGFP mouse will be a useful model for studying the relationship between alpha-synuclein aggregation and PD-specific voice and communication disorders. In its current form, this study only suggests that overexpression alone is insufficient. In contrast, a positive result in the additional experiment proposed in major comment #1 above would support this conclusion.

Response: Thank you for this thoughtful suggestion. We agree that the current results only show that alpha-synuclein overexpression is insufficient to cause vocalization changes, and do not provide evidence that alpha-synuclein pathology is required. We have tempered our conclusions to reflect this limitation of the current study.

Reviewer 2. Given that the difference was statistically significant, the authors could expand on their discussion of complex vocalizations (lines 122-128). Even if it does not mimic vocalization changes in PD patients.

Response: Thank you for this suggestion. We have added more description about this significant effect comparing it to human PD patients in the discussion section.

Reviewer 2. The introduction could include a short description of how alpha-synuclein dysregulation contributes to PD pathology. At the least, a statement regarding the relationship between alpha-synuclein overexpression and PD should be included.

Response: Thank you for this suggestion. We have added additional information about the role of alpha-synuclein overexpression and aggregation in the progression of PD and how overexpression is utilized in models of PD.

Reviewer 2. Make sure the source of all critical purchased materials is included in the methods to facilitate the reproducibility of experiments.

Response: Thank you for this suggestion. We have ensured that details for all equipment used for USV recording, identification, and analysis are included in the Methods section.

Second decision letter

MS ID#: bio.062120R1

MS TITLE: Alpha-synuclein overexpression without vocalization deficits in a mouse model of parkinsonism.

AUTHORS: Allison J Schaser; Brooke Rodgers

I am happy to tell you that your manuscript has been accepted for publication in Biology Open, pending our standard publication integrity checks. It was accepted on 4th August 2025.